# Single-pot mechanochemically-enabled fluorine atom closed-loop economy using PFASs as fluorinating agents

Hao Long[1,2], Georgina Kirby [1,2] & Lutz Ackermann [1] ✉

Per- and polyfluoroalkyl substances (PFASs), also known as "forever chemicals", pose an increasing threat to the environment and human health. Despite recent advancements in PFASs destruction, the recycling processes for such molecules remain limited to methods using high-temperatures or strong reducing agents. Sustainable degradation methods for PFASs, along with the subsequent utilization or recycling of the resulting fluorides, are indeed highly beneficial. In this study, we present a user-friendly, single-pot mechanochemical defluorination approach for fluorine transfer from PFASs to organic molecules. The high efficiency of this mechanochemical system obviates subsequent purification steps, requiring only minimal solvent filtration, even on a decagram scale. Furthermore, this strategy can be extended to the defluorination of everyday fluoroplastic and fluorinated polymers, such as PVDF membranes, pipes, and PTFE, thus addressing a critical challenge in sustainably breaking down persistent and environmentally harmful "forever chemicals".

Per- and polyfluoroalkyl substances (PFASs) have become indispensable within industries such as pharmaceuticals, electronics, energy storage, and materials[1]. Most notably, the structure of PFASs comprises of multiple strong C−F bonds, which means they have exceptional chemical stability, high-temperature resistance, low surface energy, and corrosion resistance[2-5]. However, these properties, which make them advantageous in their quotidian use, also mean that effective strategies to recycle PFASs are limited. This has then led to their widespread accumulation in water sources, soil, and within organisms, thus causing PFASs to earn the notorious label of "forever chemicals" (Fig. 1a)[6-9]. All of this leads to an increased risk to the ecosystems and to human health[10-19].

Several strategies have been explored to date for the degradation of PFASs, including photochemical[20,21], catalytic[22-24], and electrochemical decomposition (Fig. 1b)[25-28]. Kang[29] and Miyake[30] recently made significant advances in novel photocatalytic degradations of PFASs to combat this crisis. Both employed photo reductants to efficiently defluorinate and degrade a wide range of PFASs. Although

studies have been conducted in this field, significant challenges do indeed persist. This includes the difficulty in defluorinating more inert polymers and the high costs associated with the use of additives, solvents, electrolytes, and, importantly, the inefficient recovery of fluorine sources. Since fluorine is an important resource within organic chemistry, to be able to recycle it is a major driving force to finding ways to do so efficiently and sustainably[31-33]. Mechanochemical degradation (MCD) has shown great promise for polymer recycling and degradation[34-40]. However, its application to the defluorination of polymeric PFASs has thus far proven elusive[41]. We propose using MCD to efficiently cleave C−F bonds[42-44] and form new bonds[45], while also circumventing the challenges associated with the poor solubility of polymeric PFASs, such as polytetrafluoroethylene (PTFE) and polyvinylidene difluoride (PVDF).

Herein, we present a user friendly, "one-pot" approach using solely a single base in conjunction with MCD to address fluorine atom closed-loop economy via a defluorination–fluorination process (Fig. 1c). This strategy can transform a range of PFASs, including widely used fluoroplastics such as PTFE, ETFE, and fluorinated

[1]Wöhler Research Institute for Sustainable Chemistry (WISCh), Georg-August-Universität Göttingen, Tammannstraße 2, Göttingen, Germany. [2]These authors contributed equally: Hao Long, Georgina Kirby. ✉e-mail: Lutz.Ackermann@chemie.uni-goettingen.de

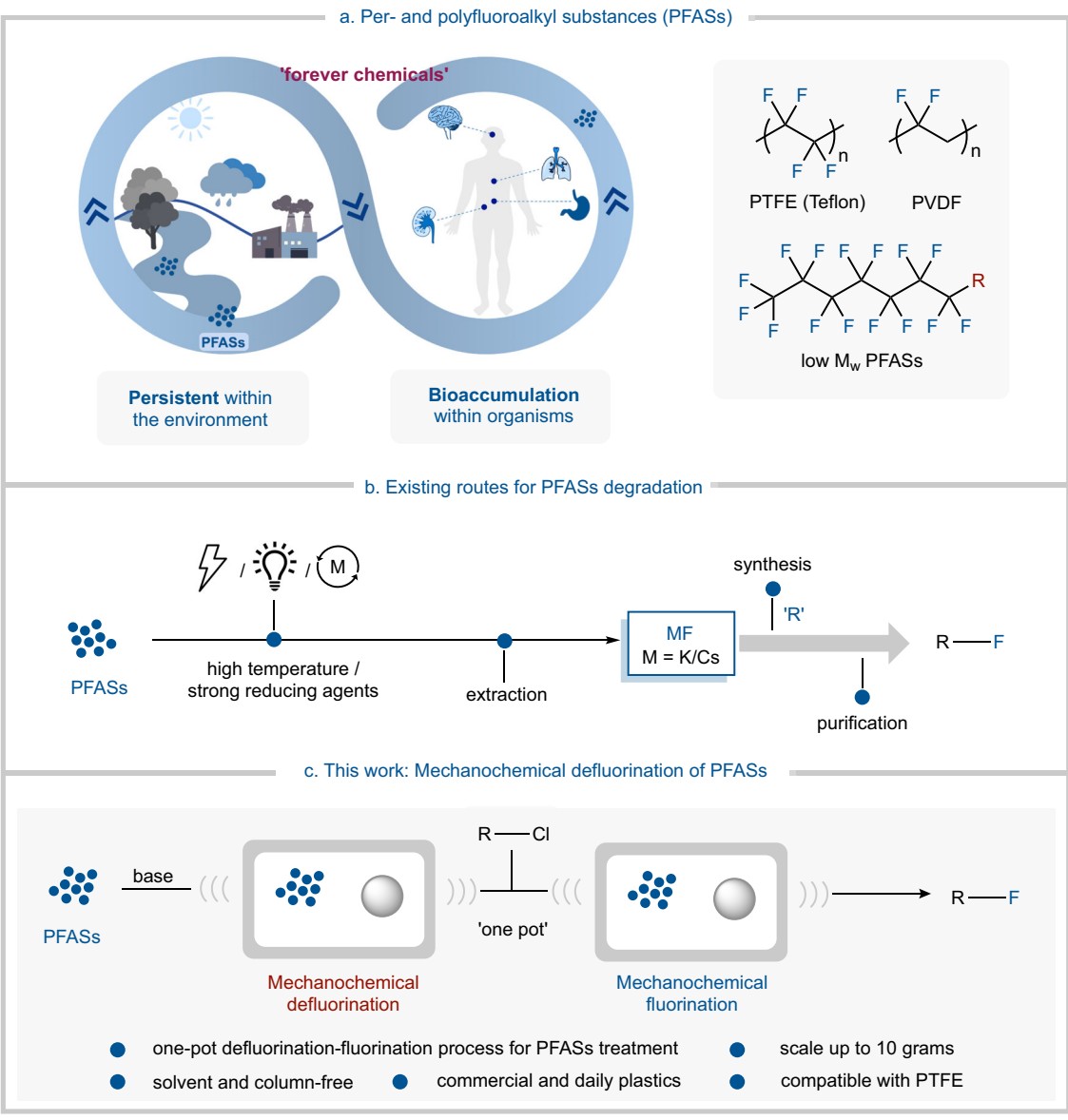

**Fig. 1 | PFASs and their degradation. a** Per- and polyfluoroalkyl substances (PFASs) and the threat to human health. **b** Existing routes for PFASs degradation. **c** This work: mechanochemical defluorination of PFASs.

ethylene propylene (FEP), to synthetically valuable sulfonyl fluorides as Sulfur (VI) Fluoride Exchange hubs (SuFEx hubs). These remarkably stable SuFEx hubs have excellent selectivity and are therefore central to the Nobel prize-winning SuFEx click chemistry[46,47]. This strategy reduces cost, waste, and energy consumption by eliminating the use of high temperatures, expensive catalysts and hazardous fluorinating agents. Additionally, our approach is scalable, allowing for the reaction to be upscaled to a 10 g scale, proving efficient for large-scale waste management. During the preparation of this manuscript, Gouverneur and Shibata published studies on the mechanochemical degradation of PFASs, enabling the isolation of fluorinating agents[48,49]. Nonetheless, compared to these approaches, our strategy has distinct advantages, as it enables both the PFASs defluorination and subsequent synthesis of high-value SuFEx precursors, in an operationally simple "single-pot" reaction without pre/post-drying of the PFAS or subsequent fluorine mixture, or column separation steps. This compares favorably to the three-step distinct steps required in the prior state-of-the-art.

## Results and discussion
### Reaction development
The aim of this strategy was to degrade and defluorinate PVDF, and to then directly perform nucleophilic fluorination in the same flask. As shown in Table 1, the ball milling was conducted in a 5 mL 304 stainless-steel jar, using a stainless-steel ball with a diameter of 7 mm. PVDF was subjected to ball milling in the presence of $t$BuOK at 30 Hz for 100 minutes (**B1**) to defluorinate the polymer. Tosyl chloride (**1**) was then introduced to this mixture, and the flask was placed back into the ball mill for an additional 100 min (**B2**). Under these conditions, only a simple filtration using $CHCl_3$ was required to obtain pure product **2** in 95% yield (Table 1, entry 1). To investigate the necessity of the base for the defluorination of PVDF, the reaction was probed without $t$BuOK, which resulted in no formation of the fluoride product (entry 2). The choice of the base significantly impacted the fluorination reaction. Inorganic bases, such as $K_3PO_4$ and $Cs_2CO_3$, resulted in moderate yields of product **2** (entries 3 and 4). However, $K_2HPO_4$, KOH and $Ca(OH)_2$ gave much lower yields (entries 3, 5 and 6). For the latter two bases,

**Table. 1 | Initial Experiments and Optimization.[a]**

| Entry | Deviation from standard condition | Yield (%)[b] |
|---|---|---|
| 1 | None | 98 (95) |
| 2 | Without tBuOK | 0 |
| 3 | $K_3PO_4$ or $K_2HPO_4$ instead of tBuOK | 49/5 |
| 4 | $Cs_2CO_3$ | 56 |
| 5 | KOH | 10 |
| 6 | $Ca(OH)_2$ | 0 |
| 7 | Pyridine | < 5 |
| 8 | 10 mm stainless steel ball | 94 |
| 9 | 15 mm $ZrO_2$ ball | 95 |
| 10 | 20 Hz, 10 Hz or 0 Hz | 20/7/0 |
| 11 | 45 min (**B1**) + 45 min (**B2**) | 80 |
| 12[c] | Reaction performed in MeCN | 5 |

[a]Reaction conditions: Ball milling was conducted in a 5 mL 304 stainless steel jar, using a 7 mm stainless-steel ball, PVDF (0.30 mmol, 1.5 equiv.), tBuOK (0.30 mmol, 1.5 equiv.), **1** (0.20 mmol), 30 Hz, 100 mins + 100 mins. [b]Yield determined by $^{19}$F-NMR analysis using 4-F-anisole as internal standard. [c]Reaction ran in round bottomed flask with 0.30 mmol of PVDF, 0.30 mmol of tBuOK and 0.20 mmol of **1** in MeCN (6 mL) for 200 mins at rt.

substantial aggregation of the material inside the jar after the initial ball milling was observed. This observation was suggestive of a possible formation of $H_2O$ from the base, thus hindering the movement of the stainless-steel ball and limiting the efficacy of the milling process. Nucleophilic fluorinating reagents are not only limited to inorganic reagents[50]. Therefore, and to explore whether this method could be extended to organic bases, pyridine was trialed. In this case, the fluorination was ineffective, giving less than 5% yield of product **2** (entry 7). In addition, the reaction proceeded well with a larger 10 mm stainless-steel ball and a $ZrO_2$ ball, giving 94% and 95% yields, respectively (entries 8 and 9). Reducing the ball milling frequency gave inferior results (entry 10), and reducing the milling time for both **B1** and **B2** to 45 minutes slightly diminished the yield (entry 11). It is noteworthy that using conventional reaction conditions, i.e., in a round-bottomed flask (RBF) in MeCN, only 5% of the fluorination product **2** was generated. This is presumably due to the low solubility of the polymer and fluoride ion in this solvent, highlighting the advantages of MCD (entry 12).

### Evaluation of substrate scope

With the optimized conditions for the fluorination of substrate **1** in hand, the robustness of the reaction was explored (Fig. 2). This sequential MCD and fluorination method showed excellent functional group tolerance (Fig. 2a). Starting with phenylsulfonyl chlorides with various substituents, ranging from electron-donating groups such as tert-butyl (**4**), methoxy (**5**), and phenyl groups (**6**) to electron-withdrawing halogens (**7–9**) and nitro groups (**10**) at different substitution positions provided excellent to near quantitative yields (90–98%). Furthermore, phenylsulfonyl chlorides containing multiple substituents were examined, including sterically hindered 2,4,6 tri-isopropyl- (**11**) and 2,4,6-trichlorophenylsulfonyl chlorides (**12**), which achieved yields of 92% and 95%, respectively. In addition, naphthyl sulfonyl chloride (**13**) gave a yield of 96%, and z−2-phenyl-ethene sulfonyl chloride (**14**) gave a yield of 90%.

Importantly, this mechanochemical fluorination demonstrated a broad tolerance for heterocyclic sulfonyl chlorides (**15–18**), with yields ranging from 73% to 91%. Notably, this method was also applicable to simple alkyl sulfonyl chlorides, such as n-dodecyl (**19**), methyl (**20**), L-(-)−10-camphorsulfonyl chloride (**21**), and cyclopropane sulfonyl chloride (**22**). Next, a variety of PFASs were evaluated. As shown in Fig. 2b, PVDF with different molecular weights were first investigated (**23–26**) using substrate **1**, all of which worked efficiently under otherwise identical conditions. This strategy also proved applicable to copolymers, with poly(vinylidene fluoride-hexafluoropropylene) (**27**) yielding product **2** with a 96% yield. It is noteworthy that the piezoelectric materials poly(vinylidene fluoride-trifluoroethylene) and poly(vinylidene fluoride-trifluoroethylene-chlorofluoroethylene) (**28, 29**) were also suitable for this reaction. In addition, defluorination of ethylene tetrafluoroethylene (**30**) gave product **2** with a 90% yield.

Interestingly, by changing the material and size of the ball mill flask to a 30 mL $ZrO_2$ flask, and using a slightly bigger 15 mm $ZrO_2$ ball, stubborn fluoroplastics and low molecular weight ($M_W$) PFASs could be defluorinated. The MCD of fluorinated ethylene propylene (**31**) generated product **2** with a 78% yield. Gratifyingly, polytetrafluoroethylene (**32**) (PTFE), which is extremely chemically and biologically inert, could also be defluorinated successfully, achieving a yield of 88% for the desired fluorinated product (Fig. 2c). Finally, with low $M_W$ PFASs, including perfluorinated heptanoic acid (**33**), sulfonyl fluoride (**34**), alkyl alcohol (**35**), olefin (**36**), and perfluorooctane (**37**), were all compatible with our mechanochemical fluorination giving yields ranging from 51% to 77% (Fig. 2d).

### Investigation of defluorination

Further investigation into the composition of the fluoride mixture after ball milling was also conducted. Using the two bases that were shown to give the high yields with PVDF for the fluorination of

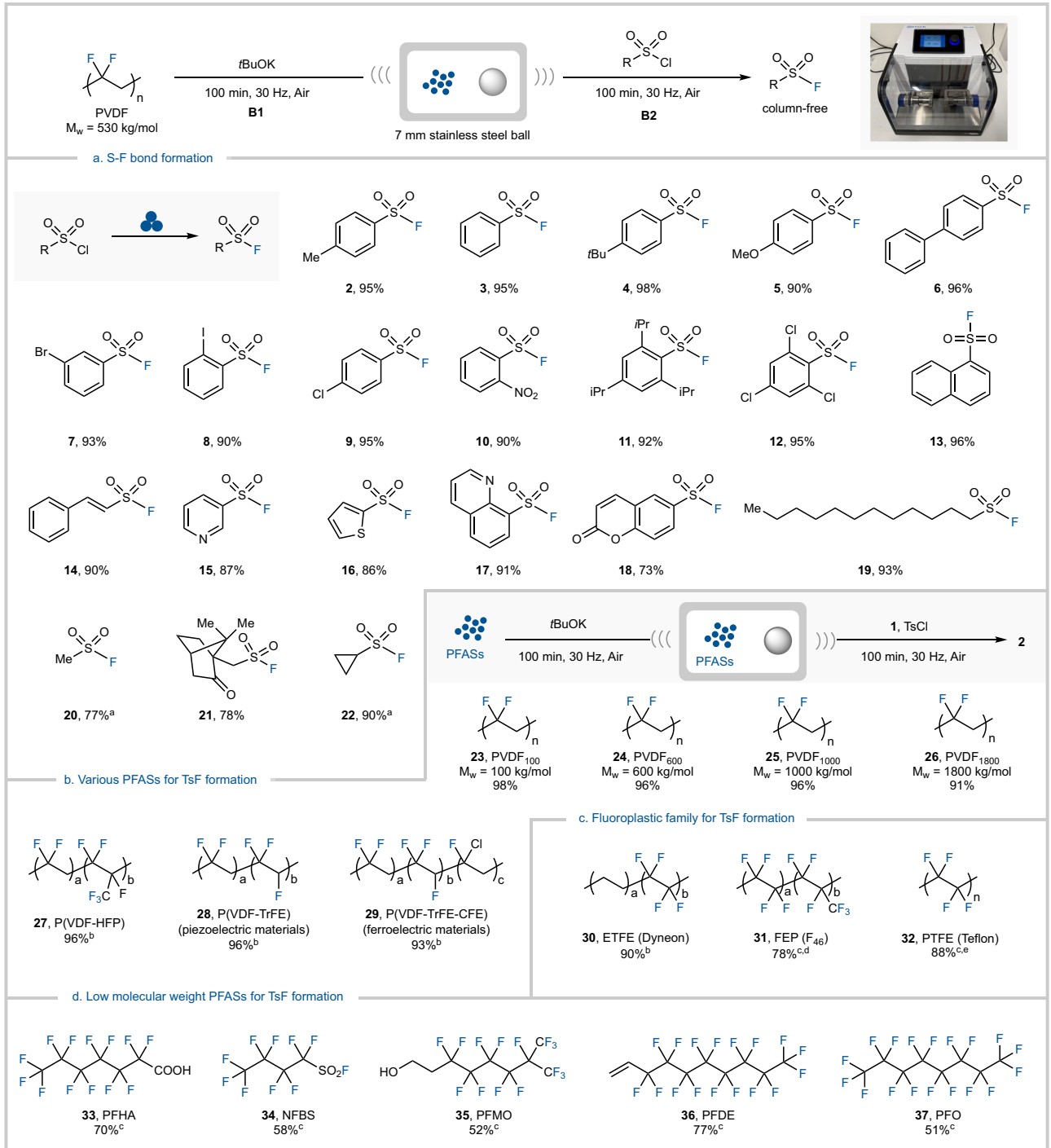

**Fig. 2 | Reaction scope.** Reaction conditions: Ball milling was conducted in a 5 mL 304 stainless-steel jar, using a stainless-steel ball with a diameter of 7 mm, PFAS (0.30 mmol, 1.5 equiv.), $t$BuOK (0.30 mmol, 1.5 equiv.), sulfonyl chloride (0.20 mmol), 30 Hz, 100 mins + 100 mins, isolated yield. [a]Yield determined by $^{19}$F-NMR analysis using 4-F-anisole as internal standard. [b]100 mg of PFAS was used. [c]Ball milling was conducted in a 30 mL ZrO$_2$ jar, using ZrO$_2$ ball with a diameter of 15 mm, PFAS (0.30 mmol, 1.5 equiv.), $t$BuOK (0.30 mmol, 1.5 equiv.), 1 (0.20 mmol), 30 Hz, 200 mins + 100 mins. [d]40 mg of PFAS used. [e]0.4 mmol, 40 mg of PTFE used. **a** Substrate scope for the formation of S-F bonds. **b** Use of various polymeric PFASs for TsF formation. **c** Use of fluoroplastics for TsF formation. **d** The use of low molecular weight PFASs for TsF formation.

substrate **1**, the generation of fluoride over time was measured using $^{19}$F-NMR, with a control experiment conducted in the absence of a base (Fig. 3a). The control exhibited no fluoride release, confirming the inherent stability of PVDF under neutral conditions. In contrast, both bases facilitated degradation, showing around 10–20% fluorine release after only 5–10 min of ball milling. Notably, the extent of defluorination increased over time, with $t$BuOK showing the highest fluoride release, suggesting a sustained and progressive defluorination process to a certain point. The differences in defluorination could stem from variations in base strength and nucleophilicity. The $^{19}$F-NMR spectrum of the PVDF and PTFE mixture in D$_2$O was measured after ball milling (Fig. 3b). In both cases, aqueous fluoride formation was observed with yields of 35% and 28%, respectively, as confirmed by comparison with KF (Fig. 3b-bottom). This demonstrates that this method indeed

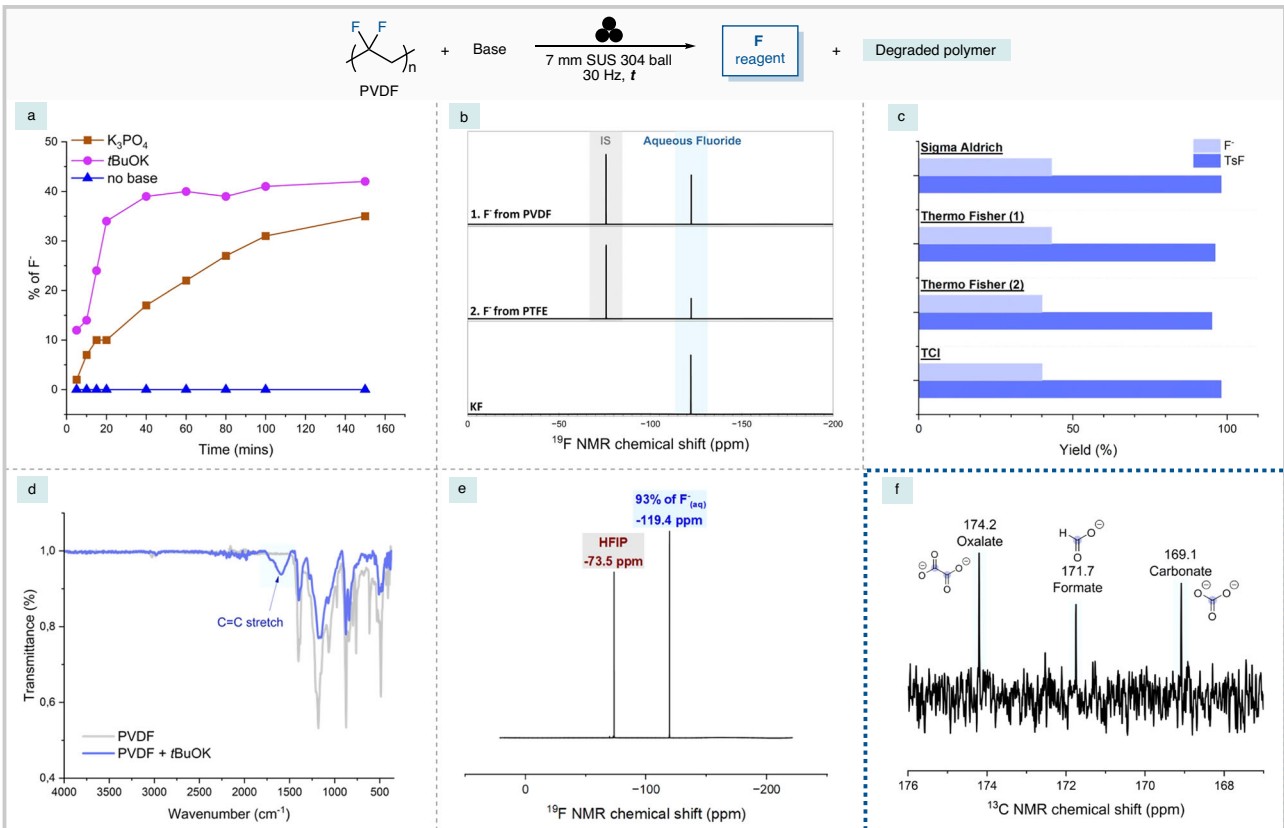

**Fig. 3 | Investigation of defluorination. a** Generation of fluoride over time from PVDF (0.3 mmol, 1 equiv.) and K$_3$PO$_4$ (0.3 mmol, 1 equiv.) and *t*BuOK (0.3 mmol, 1 equiv.). **b** Top: the $^{19}$F-NMR of the fluoride generated from MCD of PVDF and PTFE with *t*BuOK after 100 and 200 min, respectively, in D$_2$O. Bottom: the $^{19}$F-NMR of KF in D$_2$O. **c** Ball milling with different brands of *t*BuOK. **d** The FT-IR spectra of PVDF and the base-PVDF mixture after 100 mins of MCD. **e** Complete defluorination of PVDF (see SI for full details). **f** $^{13}$C-NMR of the mixture of perfluorooctane and *t*BuOK in D$_2$O. MCD = Mechanochemical degradation.

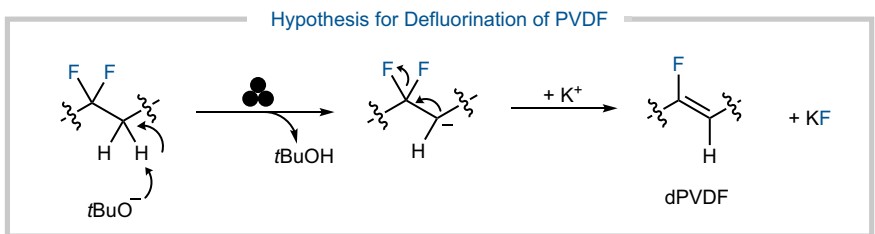

**Fig. 4 | Mechanism hypothesis.** Hypothesis for the degradation of PVDF using *t*BuOK *via* ball milling.

generates aqueous fluoride from both proton-containing and non-proton containing PFASs. To confirm the role of *t*BuOK brand variability, four different commercial sources were tested (Fig. 3c.), revealing no significant differences in fluoride yield of TsF (**2**) yield. These results further prove the robustness of this strategy. Furthermore, switching from 7 mm to 10 mm stainless steel milling balls and increasing the *t*BuOK equivalents enabled near-complete PVDF defluorination, achieving a 93% yield of fluoride (Fig. 3e.).

Infrared spectroscopy of the polymer residue after 100 minutes of ball milling provided further evidence for the degradation pathway of PVDF (Fig. 3d). The FTIR spectrum of the residue after ball milling with *t*BuOK revealed the emergence of a new C=C bond stretching peak at approximately 1600 cm$^{-1}$. This was consistent with the possible defluorination of PVDF *via* proton abstraction aided by the base. Notably, there was a significant reduction in intensity of the original PVDF vibrational peaks, which was indicative of a substantial polymer backbone modification. Surprisingly, the formation of a C≡C bond stretch peak was not observed, suggesting that the dehydro-fluorination occurred up to a certain point. With this evidence in hand, the mechanism of defluorination of PVDF could be hypothesized (Fig. 4).

This includes the deprotonation of PVDF, which then eliminates F$^-$, generating dehydrofluorinated PVDF (dPVDF) and KF. The further degradation of dPVDF is most likely much harder than the initial elimination, and may not be occurring *via* the same mechanism, hence why there is a plateau for the generation of fluoride. However, the mechanism of degradation for PTFE is most likely not going through the same path as the proposed mechanism for PVDF due to the lack of protons in the polymeric PFASs.

To gain more insight into the possible degradation pathway of PTFE, NMR studies were performed on the residual powder after ball milling. These were carried out in order to identify any small molecules formed *via* C–C bond cleavage, which were shown to occur for other methods of PTFE degradation[29]. Results from the $^1$H- and $^{13}$C-NMR spectra for the polymeric PFASs were inconclusive, due to the

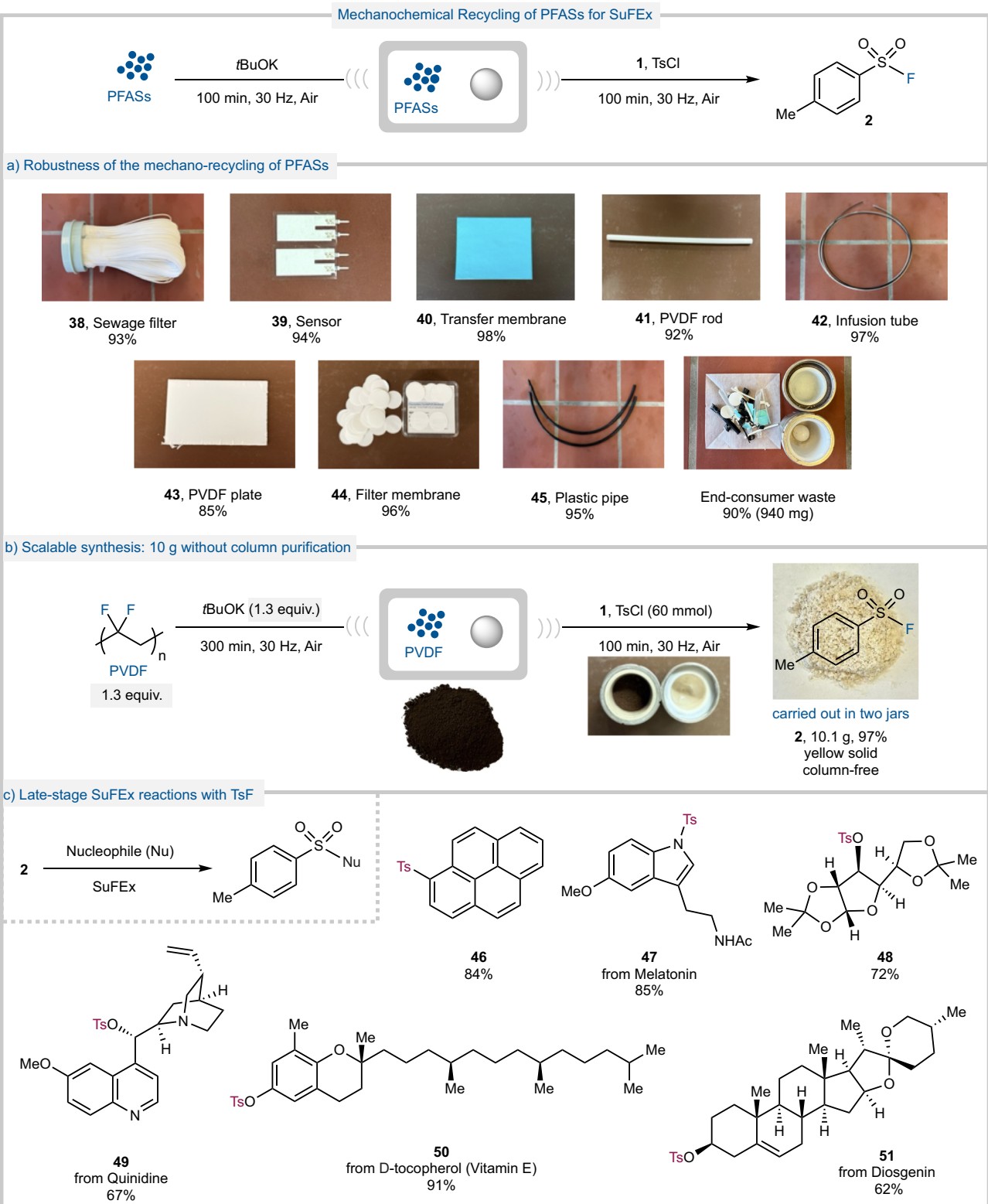

**Fig. 5 | Investigations into the robustness of the generation of SuFEx hubs. a** The showcase of the robustness of the mechanochemical recycling of PFASs. **b** 10 grams scale-up of the defluorination of PVDF. **c** Late-stage SuFEx reactions using product **2**.

formation of an insoluble black residue upon completion. This visual transformation is characteristic of the formation of amorphous carbon (char). To gain further mechanistic insight, the $^{13}$C-NMR of the mixture obtained from the ball milling of perfluorooctane with *t*BuOK was performed. This showed trace formation of oxalate (174.2 ppm), formate (171.7 ppm) and carbonate (169.1 ppm) within the sample (Fig. 3f).

This confirms that a portion of the carbon content for perfluorooctane undergoes oxidative mineralization to small, water-soluble anions. This could give an indication of the mechanism of degradation for PFASs with no protons, which could proceed through a radical pathway, whereby there is defluorination and consequent C–C bond cleavage[51–53]. However, the pathway involving an $S_N2$ type

defluorination cannot be ruled out, where there is direct fluoride displacement and base mediated bond scission[48].

## Transformation and application

PFASs are ubiquitous in daily life and are seen in many everyday products, thus, they are often discarded after a single use, contributing to environmental pollution. To evaluate the practicality of our MCD method, we tested the defluorination reaction on various PFASs-containing waste materials commonly encountered in daily life (Fig. 5a). These included items such as filter membranes (**38, 40, 44**), tubing (**42, 45**) and sensor materials (**39**). Notably, these PFASs substrates achieved nearly quantitative yields in the fluorination of TsCl, generating TsF (**2**) without requiring chromatographic purification.

Crucially, the method proved highly scalable, maintaining efficiency even at the 10 g scale (Fig. 5b). This scale-up was achieved using two parallel jars and required only 1.3 equivalents of PVDF and base, with no compromise in the yield or purity of the TsF product (**2**). A mixed waste stream containing 1 gram of blended PVDF-derived waste (**38–45**) was also successfully processed, producing 940 mg of TsF (**2**) (90% yield). The products from both large-scale reactions were isolated simply *via* filtration using a minimal amount solvent, highlighting our method's industrial viability and sustainability. Sulfonyl fluorides constitute the core electrophiles of SuFEx chemistry, one of the next-generation click reactions pioneered by Sharpless and coworkers. Such reactions have been recognized by the 2022 Nobel Prize in Chemistry for its robust, modular, and biocompatible bond-forming potential[46,47,54,55]. Given the importance of sulfonyl fluorides in SuFEx click chemistry, the reactivity of the synthesized TsF (**2**) was explored through a series of late-stage SuFEx transformations (Fig. 5c). The formation of an S−C bond was demonstrated by coupling **2** with a fluorescent aromatic substrate, affording compound **46** in excellent yield. Starting from Melatonin, tosylation of the pyrrole nitrogen yielded the N−S bonded product **47** in 85% yield. A range of alcohols were also subjected to SuFEx reactions with TsF (**2**). The tosylation of the hydroxyl group in diacetone-*d*-glucose afforded the sulfonate ester **48** in 72% yield. Similarly, the reaction with the antimalarial drug Quinidine afforded compound **49** in 67 % yield. The SuFEx-mediated tosylation of δ-tocopherol (form of vitamin E) afforded the tosylated product **50** in an excellent 91 % yield. Finally, the reaction with the natural steroid, Diosgenin, yielded the tosylated product **51** in 62% yield. These transformations showcase the broad synthetic utility of the generated TsF (**2**) in generating valuable sulfonyl derivatives across multiple classes of compounds. By demonstrating that fluorine atoms can be reclaimed from plastic waste and upcycled into functional $SO_2F$ motifs, our method not only addresses a pressing environmental challenge but also provides direct access to click-enabled materials, expanding the synthetic utility of SuFEx chemistry.

To conclude, we have developed a sustainable and operationally simple single-pot mechanochemical defluorination of PFASs, including highly inert fluoropolymers such as PVDF and PTFE. This user-friendly method requires a base and a ball mill, and enables the direct transfer of fluorine atoms from PFASs to synthetically valuable sulfonyl fluorides, without the need for harsh reagents, high temperatures or extensive purification. The reaction is scalable, waste-tolerant and compatible with a wide variety of PFASs ranging from low Mw PFASs to consumer waste materials. Crucially, this method provides direct access to a SuFEx hub in high yields, offering a valuable route towards a closed-loop fluorine economy. This work also addresses the challenge in recycling these "forever chemicals", thus, bringing together environmental remediation with synthetic utility[7].

## Methods

### General procedure for mechanochemical defluorination of PFASs

Ball milling reactions were performed under air. The reagents were used without pre-purification or drying. To a 5 mL Reich stainless-steel milling jar equipped with a PTFE sealing ring was added a 7 mm stainless-steel ball, PFASs (1.5 equiv.), and *t*BuOK (1.5 equiv.). The jar was tightly sealed, mounted on the shaker mill, and milled at 30 Hz for a milling cycle (**B1**). Upon completion of **B1**, the jar was opened, and sulfonyl chloride (1.0 equiv.) was added to the resulting black mixture. The jar was then resealed and subjected to a second milling cycle (**B2**). After completion of **B2**, 3 mL of $CHCl_3$ was added to wash the black solid. The mixture was filtered through a small pad of celite®, and the reaction vessel was subsequently rinsed with an additional $2 \times 3$ mL of $CHCl_3$, followed by a second filtration. The combined filtrates were concentrated under reduced pressure and dried under vacuum to afford the pure target product without the need for further purification. The substrates were either purchased directly from commercial suppliers or prepared according to previously reported procedures (see supplementary information). All other reagents and solvents used in this study were purchased from commercial sources and used as received.

## Data availability

The authors declare that the data supporting the findings of this study are available within the paper and its supplementary Information files. All other requests for materials and information should be addressed to the corresponding authors.

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

## Acknowledgements

The authors gratefully acknowledge support from the ERC Advanced Grant No. 101021358, the DFG (Gottfried Wilhelm Leibniz award to L.A.). The authors also thank Dr. Michael John for help with NMR studies, S. Beusshausen for the technical assistance and S. Trienes for the ICP-MS measurements. The authors also thank F. Gerlich for help with corrections. A preprint of this publication is available on ChemRxiv (DOI:10.26434/chemrxiv-2025-02hh7).

## Author contributions

H.L., G.K., and L.A. conceived and directed the project. H.L. and G.K. performed the experiments. All authors contributed to data analysis. The manuscript was prepared with contributions from all authors.

## Funding

## Competing interests

The authors declare no competing interests.
