## [Transparent Peer Review file · Nature Communications]

Single-Pot Mechanochemically-Enabled Fluorine Atom Closed-Loop Economy Using PFASs as Fluorinating Agents

Corresponding Author: Professor Lutz Ackermann

Version 0:

Reviewer comments:

Reviewer #1

(Remarks to the Author)

Long and coworkers describe a mechanochemical strategy employing KOtBu as a base for the defluorination of fluorinated polymers and other PFASs. The method is noteworthy for its capacity to synthesize high-value SuFEx precursors in a single pot with high yield and minimal postprocessing. Nonetheless, several technical and interpretive issues should be addressed prior to publication.

1. The substrates 31 and 34–36 are labeled with subscript “b.” According to the figure legend, this subscript indicates PVDF. This annotation appears inconsistent with the described chemical structures and should be clarified.
2. A mechanistic explanation should be provided to rationalize why K_3PO_4 yields superior results compared with tBuOK, despite tBuOK being both a stronger base and nucleophile. Discussion in the context of Gouverneur’s system would improve clarity.
3. The dosage of the internal standard (HFIP) or the exact formula used for calculating the fluorine yield should be included. The fluorine yield in Figure 3c must be explicitly reported. In Figure S8, if the integral “0.37” corresponds to 0.37 mmol fluoride using 0.3 mmol tBuOK, the source of the additional 0.07 mmol should be clarified. Additionally, the relative peak intensities at 122.4 ppm in Figures S8 and S9 are inconsistent with the stated integrals; this discrepancy should be examined.
4. The reason for lower yields with low-molecular-weight PFASs (particularly 35 and 37) should be discussed. Possible factors include volatility, low melting point, or inherent difficulty of defluorination.
5. The brand and quality of tBuOK, should be specified. Since results may vary among suppliers, testing tBuOK from different manufacturers is recommended to ensure reproducibility.
6. For PFAS-containing waste materials, it should be verified whether additives or dyes (especially in black plastic pipes) affect the purity of TsF when only $CHCl_3$ extraction is used. 1H NMR spectra should be provided to confirm product identity and purity.
7. The destination of the remaining fluorinated polymeric residue should be considered. Discussion should include whether such material poses environmental risks.
8. Since generating TsF from PVDF and KOtBu is costlier than direct methods (e.g., using KF), this limitation should be acknowledged.
9. It is recommended to report the defluorination yield relative to the total fluorine content of the polymer, rather than only the yield of the small-molecule product. If possible, the authors should provide an example demonstrating complete defluorination, for example using excess potassium base, to illustrate that no partially fluorinated polymeric byproducts are formed, thereby avoiding potential new environmental concerns.

Reviewer #2

(Remarks to the Author)

This is an interesting study on the mechanochemical degradation of PFAS polymers. The resulting fluorides are subsequently used for fluorination to obtain sulfonyl fluorides, which are SuFEx precursors. The manuscript should not be published in Nature communications. I suggest publication in a specialized journal.

Recently, two articles with very similar content were published recently by Shibata and Gouverneur (Ref 48,49). They essentially describe analogous results. In both papers PFAS polymers were defluorinated mechanochemically and the

obtained fluorides were used for fluorination. The formation of sulfonyl fluorides and other S(VI) compounds was also described. Additionally, in the paper by Shibata the same strategy was used for defluorination, e.g. addition of tBuOK to the ball milling process to give KF.

Mechanistically the paper by Ackermann and coworkers suggests for the PVDF degradation the same pathways as published before. There are no conclusive mechanistic studies for the PTFE defluorination and the authors suggest “a radical pathway”, which, however, should be further investigated, for instance by trapping experiments. It is not clear what happens to the carbon content of the polymers. The authors mention the formation of “traces of oxalate, formate and carbonate”, but they did not study this further. This should be done.

Version 1:

Reviewer comments:

Reviewer #1

(Remarks to the Author)

This manuscript has been well revised. Although the authors have made significant efforts to address some of the comments by the authors, there is one improvement necessary for publication.

- On Page 6, Line 111, the author described that 0.3 mmol of PFAS was applied. However, the amount for some substrates seems incorrect. For example, on Page 29 of the SI, 40 mg of PTFE was used, which corresponds to 0.4 mmol when C₂F₄ is considered as the structural unit. The author should check this carefully. It would be helpful if the author could provide the yield calculation formula for the substrate 30 to 33.

Reviewer #2

(Remarks to the Author)

The authors made a good effort to address the questions raised by Reviewer 1. However, I still think that the paper is not at all suitable for Nature Communications. Very related results have been published before by Shibata and Gouverneur. I mentioned this before and like to emphasize again that “they essentially describe analogous results. In both papers PFAS polymers were defluorinated mechanochemically and the obtained fluorides were used for fluorination. The formation of sulfonyl fluorides and other S(VI) compounds was also described. Additionally, in the paper by Shibata the same strategy was used for defluorination, e.g. addition of tBuOK to the ball milling process to give KF.” KF is also formed in the manuscript by Ackermann.

The authors claim in their rebuttal letter that the “developed strategy differs from these reports in operational simplicity, product focus, and mechanistic insights, as summarized in the table below.” Essentially, the Shibata method were simply optimized and altered (see Table on the rebuttal letter) to obtain similar fluorinated products. Mechanistic studies also relate to the Gouverneur paper and a paper by Will Dichtel. The new methods that have been applied/developed are very minor, at least when the results should be published in Nature Communications.

In addition, a preprint published online before the Shibata publication does not justify publication in a prestigious journal., as suggested by the authors.

The studies requested by the authors concerning the black residue after ball milling were inconclusive. I quote them: “Results from the 1H- and 13C-NMR spectra for the polymeric PFASs were inconclusive, due to the formation of an insoluble black residue upon completion.”

Responses to reviewers

Reviewer #1 (Remarks to the Author): Long and coworkers describe a mechanochemical strategy employing KOtBu as a base for the defluorination of fluorinated polymers and other PFASs. The method is noteworthy for its capacity to synthesize high-value SuFEx precursors in a single pot with high yield and minimal postprocessing. Nonetheless, several technical and interpretive issues should be addressed prior to publication.

Comment 1.: The substrates 31 and 34–36 are labelled with subscript “b.” According to the figure legend, this subscript indicates PVDF. This annotation appears inconsistent with the described chemical structures and should be clarified.

Response: We thank the reviewer for the comment and suggestion. The figure legend has now been changed accordingly.

Comment 2.: A mechanistic explanation should be provided to rationalize why K₃PO₄ yields superior results compared with tBuOK, despite tBuOK being both a stronger base and nucleophile. Discussion in the context of Gouverneur’s system would improve clarity.

Response: We thank the reviewer for this comment. In our case, tBuOK yields superior results than with K₃PO₄. As we envisioned the development of a user-friendly approach to defluorination/fluorination, neither our base nor the PFAS used were pre-treated. The Gouverneur group disclosed that anhydrous K₃PO₄ was necessary to achieve high defluorination yields, which accounts for why it is not working as well in our system. Additionally, the fluorination using K₃PO₄ in our strategy yielded 49% of TsF (Table 1 entry 3 in the main text), which we believe is due to the lower nucleophilicity of the PFAS/base mixture generated in B1, in comparison to our tBuOK/PFAS mixture.

Comment 3.: The dosage of the internal standard (HFIP) or the exact formula used for calculating the fluorine yield should be included. The fluorine yield in Figure 3c must be explicitly reported. In Figure S8, if the integral “0.37” corresponds to 0.37 mmol fluoride using 0.3 mmol tBuOK, the source of the additional 0.07 mmol should be clarified. Additionally, the relative peak intensities at 122.4 ppm in Figures S8 and S9 are inconsistent with the stated integrals; this discrepancy should be examined.

Response: We thank the reviewer for this comment. For the fluoride yield 10 μL of HFIP is used as an internal standard. The internal standard is always integrated for 1. The formulas used are as follows:

For HFIP as an internal standard:

$$\text{Moles of product} = \left(\frac{\text{Integral of Product}}{(\text{Integral of Internal standard})/6} \right) \times \text{Moles of Internal standard}$$

$$\text{Yield of fluorine} = \frac{\left(\frac{\text{moles of product}}{\text{limiting reagent moles}} \times 100 \right)}{\text{number of fluorines in PFAS}}$$

For example, for Figure S8, the moles of product are

$$\text{Moles of product} = \left(\frac{0.37}{1/6}\right) \times 0.0095 = 0.2109 \text{ mmol}$$

Thus,

$$\text{Yield of fluorine} = \frac{\left(\frac{0.2109}{0.3} \times 100\right)}{2} = 35.15 \%$$

The formula for the fluorine yield determination was added to the revised Supporting Information (page S3). As suggested, a discussion regarding the yield in the now Figure 3b was added to the revised manuscript as follows:

“The ^{19}F -NMR in D_2O of the PVDF and PTFE mixture was measured after ball milling (Fig. 3b). In both cases, the formation of aqueous fluoride was observed with a yield of 35 % and 28 %, respectively, and was confirmed by comparison with KF (Fig. 3b-bottom).”

In terms of the intensities of the peaks, this is most likely to do with the dilution/concentration of the sample taken for the ^{19}F -NMR.

Comment 4.: The reason for lower yields with low-molecular-weight PFASs (particularly 35 and 37) should be discussed. Possible factors include volatility, low melting point, or inherent difficulty of defluorination.

Response: We thank the reviewer for this comment. The diminished efficiency observed for low molecular weight PFASs most likely arises from multiple related factors, including low volatility or low melting points, and the resistance of certain low molecular weight PFAS to defluorination. Additionally, the C-F bonds may be less accessible in comparison to the solid polymeric PFASs. For example, the milling mixture for liquid low molecular weight PFASs (e.g. PFOS, perfluorooctane) not being homogeneous and therefore does not mix as efficiently.

Comment 5.: The brand and quality of *t*BuOK, should be specified. Since results may vary among suppliers, testing *t*BuOK from different manufacturers is recommended to ensure reproducibility.

Response: We thank the reviewer for this suggestion. The brand of *t*BuOK used throughout the manuscript is from TCI. We have since trialed the defluorination of PVDF with *t*BuOK from four different suppliers (Figure 3). There is no significant difference between each brand of bases. The results were as shown:

Entry	t BuOK from different manufacturers/batches	Fluoride yield(%) ^b
1	97% pure, TCI, P1008	38%
2	98% pure, Thermo Fisher, 168881000 (25g)	43%
3	98% pure, Thermo Fisher, 168885000 (500g)	39%
4	98% pure, Sigma-Aldrich, 156671	43%

In addition, ICP-MS analysis of the *t*BuOK was performed for each brand in order to verify their respective composition. This analysis showed no significant amount of metals in each sample of the base. After ball milling in the stainless steel 5mL jar with a 7mm stainless steel ball for 100 minutes at 30 Hz, there is, as expected, an increase of certain metals, in this case mostly iron. These results have since been added to the Supporting Information (page S12).

ICP-MS data									
KO t -Bu	Mn	Ni	Cu	Cr	Fe	Pd	Ir	Co	Zn
	(ppm)	(ppm)	(ppm)	(ppm)	(ppm)	(ppm)	(ppm)	(ppm)	(ppm)
TCI	0.21	0.34	4.82	2.07	13.6	0.006	0.004	0.06	2.26
Thermo Fisher (500 g)	0.25	1.57	5.52	2.17	22.4	0.004	-	0.08	2.34
Thermo Fisher 25g	0.22	1.82	4.09	1.97	17.8	0.007	-	0.06	2.39
Sigma Aldrich	0.21	0.29	3.05	3.39	15.9	0.011	-	0.04	1.49
TCI (after ball milling)	4.58	0.33	16.6	20.3	129.3	0.489	-	0.58	48.2

- = intensity below calibration range

Comment 6.: For PFAS-containing waste materials, it should be verified whether additives or dyes (especially in black plastic pipes) affect the purity of TsF when only CHCl₃ extraction is used. ¹H NMR spectra should be provided to confirm product identity and purity.

Response: We thank the reviewer for this comment. For all PFAS-containing waste materials the purity of the TsF is >95 % when only CHCl₃ is used, where only the *t*BuOH from the base is seen in the crude NMR, which is removed under vacuum when drying the TsF. The crude NMR spectra for each reaction (Figure 5, **38-45**) were added to the revised Supporting Information.

Comment 7.: The destination of the remaining fluorinated polymeric residue should be considered. Discussion should include whether such material poses environmental risks.

Response: We thank the reviewer for highlighting this aspect. All remaining fluorinated PFAS residue was disposed of through standard laboratory hazardous waste protocols at our institution, in full compliance with German laboratory safety regulations (TRGS 526) and EU waste handling guidelines. These protocols ensure proper segregation, collection by certified waste services, and thermal treatment to prevent any environmental leaching or waterway contamination. This, therefore, eliminates the environmental risks from residual materials. We have added a statement confirming these procedures to the Experimental Section (Supporting Information page S3).

Comment 8.: Since generating TsF from PVDF and KO*t*Bu is costlier than direct methods (e.g., using KF), this limitation should be acknowledged.

Response: We thank the reviewer for this important point. The generation of TsF from PVDF in the presence of *t*BuOK involves more steps compared to direct fluorination approached using KF, which may make it slightly costlier. However, the primary aim of our developed strategy is to establish the use of waste PFAS materials, which are environmentally persistent and harmful “forever chemicals”, as a sustainable fluorine source. Our approach valorizes problematic PFAS waste by converting it into

valuable organic fluorinated compounds, addressing urgent environmental and health concerns by both degrading persistent pollutants and recycling their fluorine content.

Comment 9.: It is recommended to report the defluorination yield relative to the total fluorine content of the polymer, rather than only the yield of the small-molecule product. If possible, the authors should provide an example demonstrating complete defluorination, for example, using excess potassium base, to illustrate that no partially fluorinated polymeric byproducts are formed, thereby avoiding potential new environmental concerns.

Response: We thank the reviewer for this comment and suggestion. We explored various reaction conditions to completely defluorinate PVDF. Thus, we were able to show near complete defluorination by running the ball mill firstly with two equivalents of *t*BuOK for 100 minutes using a larger 10mm stainless steel milling ball, followed by the addition of another 2 equivalents of *t*BuOK for another 100 minutes of ball milling, yielding 93 % of fluorine. These results were added to the revised Supporting Information (page S15) and manuscript (figure 3).

In this context, an updated discussion was added to the revised manuscript as follows: “Furthermore, switching from 7 mm to 10 mm stainless steel milling balls and increasing the *t*BuOK equivalents enabled near-complete PVDF defluorination, achieving a 93% yield of fluoride (**Fig 3e.**)”

Reviewer #2 This is an interesting study on the mechanochemical degradation of PFAS polymers. The resulting fluorides are subsequently used for fluorination to obtain sulfonyl fluorides, which are SuFEx precursors. The manuscript should not be published in Nature communications. I suggest publication in a specialized journal.

Comment 1: Recently, two articles with very similar content were published recently by Shibata and Gouverneur (Ref 48,49). They essentially describe analogous results. In both papers PFAS polymers were defluorinated mechanochemically and the obtained fluorides were used for fluorination. The formation of sulfonyl fluorides and other S(VI) compounds was also described. Additionally, in the paper by Shibata the same strategy was used for defluorination, e.g. addition of *t*BuOK to the ball milling process to give KF.

Response: We thank the reviewer for highlighting these important recent studies by Shibata and Gouverneur. Our preprint was publicly available on ChemRxiv prior to the Shibata publication ([10.26434/chemrxiv-2025-02hh7](https://doi.org/10.26434/chemrxiv-2025-02hh7)), establishing the independent development of this approach. Nonetheless, we fully acknowledge both contributions and consider our work complementary. Our developed strategy differs from these reports in operational simplicity, product focus, and mechanistic insights, as summarized in the table below.

First, our approach operates under markedly simpler conditions: both *t*BuOK and the PFAS substrates are used directly as received, without any pre-drying or pre-purification. Second, our approach uniquely delivers high-value sulfonyl fluorides (SuFEx hubs) in a true one-pot fashion, directly coupling PFAS defluorination with synthetically useful fluorination in a single operational sequence. Finally, our mechanistic analysis supports both an elimination and nucleophilic (S_N2 -type)

pathway under *t*BuOK-mediated mechanochemical conditions, whereas the Gouverneur study is consistent with a purely S_N2 pathway and the Shibata group with a dehydrofluorination-type mechanism. We have clarified these distinctions in the revised manuscript and hope this makes the complementary nature of our contribution more apparent.

Criteria	This Work (Ackermann)	Gouverneur et al.	Shibata et al.
Base	t BuOK (used as received)	K ₃ PO ₄ or K ₄ P ₂ O ₇ (must be anhydrous)	t BuOK
Pretreatment of PFAS	None	None	Requires drying at 120°C under vacuum
Solvent use	solvent-free mechanochemistry	solvent-free mechanochemistry	Liquid-assisted grinding (THF LAG, 0.5 μL/mg)
After milling	Simple filtration only	Requires extraction, precipitation, chromatography	Requires vacuum drying at 120°C
Mechanism	Predominantly base mediated elimination and S _N 2 (depending on PFAS)	S _N 2 nucleophilic substitution	Base-mediated elimination
Product	High-value sulfonyl fluoride SuFEx hubs (proven use by late stage SuFEx reactions)	Complex mixtures (KF, K ₂ PO ₃ F, carbonates)	KF. Black intermediate requiring further processing
Single-Pot Reaction	Yes , defluorination and fluorination in one pot	No , requires separate steps	No , requires separate drying and fluorination steps

Comment 2: Mechanistically the paper by Ackermann and coworkers suggests for the PVDF degradation the same pathways as published before. There are no conclusive mechanistic studies for the PTFE defluorination and the authors suggest “a radical pathway”, which, however, should be further investigated, for instance by trapping experiments.

Response: We thank the reviewer for this comment. We have since changed the manuscript regarding the ‘radical pathway’. The evidence for the radical pathway for low molecular weight PFASs is currently still ongoing within the lab. However, we believe that the defluorination of PTFE and perfluorooctane under our conditions ball milling with *t*BuOK, likely proceeds through either a base-mediated elimination, S_N2 pathway combined with possible radical formation. This supports why our strategy presents a broad substrate scope for different types of PFASs.

S_N2 pathway: The Gouverneur group demonstrated that phosphate-mediated PTFE degradation proceeds via an S_N2 mechanism, which they calculated via DFT (*Nature* **2025**, *640*, 100–106). The strong nucleophilicity and basicity of *t*BuOK in our system mean that it is equally, or more capable of nucleophilic attack of the C–F bonds in PTFE, perfluorooctane, and other similar PFASs. This can enable direct fluoride displacement, and base mediates C–C bond scission.

Ball milling is well-known to induce C–C bond cleavage in polymers, generating reactive macroradicals (Briš *et al.*, *Green Chem.* **2025**, *27*, 11345–11382). Critically, this can be achieved using both stainless-steel and ZrO₂ milling flasks and balls. Furthermore, for our strategy, we were able to identify the formation of carbonates, formates, and oxalates from the ball milling of perfluorooctane, which is similar to the work from Kang (*Nature* **2024**, *635*, 610–617) who suggested a similar radical-mediated carbon backbone fragmentation.

In this context, we included a discussion in the revised manuscript as follows:

“To gain more insight into the possible degradation pathway of PTFE, NMR studies were performed on the residual powder after ball milling. This was done to identify any small molecules formed *via* C–C bond cleavage which has been shown to occur for other methods of PTFE degradation.²⁹ Results from the ¹H- and ¹³C-NMR spectra for the polymeric PFASs were inconclusive, due to the formation of an insoluble black residue upon completion. This visual transformation is characteristic of the formation of amorphous carbon (char). To gain further mechanistic insight, the ¹³C-NMR of the mixture obtained from the ball milling of perfluorooctane with *t*BuOK was performed. This showed trace formation of oxalate (174.2 ppm), formate (171.7 ppm), and carbonate (169.1 ppm) within the sample (**Fig. 3f**). This confirms that a portion of the carbon content for perfluorooctane undergoes oxidative mineralization to small, water-soluble anions. This could give an indication as for the mechanism of degradation for PFASs with no protons, which could proceed through a radical pathway, whereby there is defluorination and consequent C–C bond cleavage.⁵¹⁻⁵³ However, the pathway involving an S_N2 type defluorination cannot be ruled out, where there is direct fluoride displacement and base mediated bond scission.⁴⁸”

Comment 3: It is not clear what happens to the carbon content of the polymers. The authors mention the formation of “traces of oxalate, formate, and carbonate”, but they did not study this further. This should be done.

Response: We thank the reviewer for this comment regarding the fate of the carbon content. As discussed in the manuscript (and shown in Figure 3d), the degradation of the model substrate perfluorooctane yields soluble mineralized species, specifically formate, oxalate, and carbonate, which we identified *via* ¹³C-NMR.

For the polymeric PFASs (PVDF/PTFE), we observed the formation of a significant amount of an insoluble black solid during the reaction. This is characteristic of the formation of amorphous carbon/char, which is frequently observed in mechanochemical and photocatalytic defluorination (e.g., Shibata, *Nature* **2024**, *635*, 610; *Nature* **2025** *640*, 100–106), and the presence of remaining partially- and completely defluorinated polymers. In this regard, we have now updated the discussion in the "Investigation of Defluorination" section as follows: “To gain more insight into the possible degradation pathway of PTFE, NMR studies were performed on the residual powder after ball milling. These were carried out in order to identify any small molecules formed *via* C–C bond cleavage, which were shown to occur for other methods of PTFE degradation.²⁹ Results from the ¹H- and ¹³C-NMR spectra for the polymeric PFASs were inconclusive, due to the formation of an insoluble black residue upon completion.”

Responses to reviewers

Reviewer #1 (Remarks to the Author):

This manuscript has been well revised. Although the authors have made significant efforts to address some of the comments by the authors, there is one improvement necessary for publication.

Comment: On Page 6, Line 111, the author described that 0.3 mmol of PFAS was applied. However, the amount for some substrates seems incorrect. For example, on Page 29 of the SI, 40 mg of PTFE was used, which corresponds to 0.4 mmol when C₂F₄ is considered as the structural unit. The author should check this carefully. It would be helpful if the author could provide the yield calculation formula for the substrate 30 to 33.

Response: We thank the reviewer for this observation. For co-polymers **27-31**, an excess of PFAS was used (**27-30** = 100 mg, **31** = 40 mg) due to a variation of the ratio of the co-polymer segments. This was carried out to ensure that effective defluorination could be achieved for these polymers. For PTFE (**32**), a slight excess of 2 equiv. (0.4 mmol) was used. This was rectified in the revised manuscript (Fig. 2 caption) and the revised Supporting Information.

Reviewer #2 (Remarks to the Author):

Comment: The authors made a good effort to address the questions raised by Reviewer 1. However, I still think that the paper is not at all suitable for Nature Communications. Very related results have been published before by Shibata and Gouverneur. I mentioned this before and like to emphasize again that “they essentially describe analogous results. In both papers PFAS polymers were defluorinated mechanochemically and the obtained fluorides were used for fluorination. The formation of sulfonyl fluorides and other S(VI) compounds was also described. Additionally, in the paper by Shibata the same strategy was used for defluorination, e.g. addition of tBuOK to the ball milling process to give KF.” KF is also formed in the manuscript by Ackermann. The authors claim in their rebuttal letter that the “developed strategy differs from these reports in operational simplicity, product focus, and mechanistic insights, as summarized in the table below.” Essentially, the Shibata method were simply optimized and altered (see Table on the rebuttal letter) to obtain similar fluorinated products. Mechanistic studies also relate to the Gouverneur paper and a paper by Will Dichtel. The new methods that have been applied/developed are very minor, at least when the results should be published in Nature Communications. In addition, a preprint published online before the Shibata publication does not justify publication in a prestigious journal., as suggested by the authors.

The studies requested by the authors concerning the black residue after ball milling were inconclusive. I quote them: “Results from the 1H- and 13C-NMR spectra for the polymeric PFASs were inconclusive, due to the formation of an insoluble black residue upon completion.”

Response: Thank you for your continued feedback and for acknowledging our efforts in addressing the prior comments. We appreciate your emphasis on the contributions

by Shibata, Gouverneur, which we have indeed cited. We respectfully stand by the differences outlined in our previous reviewer response, which we believe provides support for the publication of our manuscript in *Nature Communications*. Regarding the preprint, it was mentioned for chronological context and transparency, as our strategy was already developed before the publication of Shibata and coworkers, and remained unchanged. Further studies into the residual PFAS black powder and the exact mechanism for PFASs containing no hydrogens (e.g. PTFE, PFO) are currently ongoing within our research group. We apologize if this wasn't made clear in the previous reviewer response.